Advances in research on the main nutritional quality of daylily, an important flower vegetable of Liliaceae

Wang Li-Xiang 1 2
Wang Ya-Hui 1 3
Chen Chen 1 2 3
Liu Jie-Xia 1 3
Li Tong 1 3
Li Jing-Wen 1 3
Liu Pei-Zhuo 1 3
Xu De-Bao 1 3
Shu Sheng 1 2 shusheng@njau.edu.cn
http://orcid.org/0000-0002-7900-5001 Xiong Ai-Sheng 1 2 3 xiongaisheng@njau.edu.cn
1 Suqian Research Institute, College of Horticulture, Nanjing Agricultural University , Nanjing, Jiangsu , China
2 Facility Horticulture Research Institute of Suqian , Suqian, Jiangsu , China
3 State Key Laboratory of Crop Genetics and Germplasm Enhancement and Utilization/Ministry of Agriculture and Rural Affairs Key Laboratory of Biology and Germplasm Enhancement of Horticultural Crops in East China, Nanjing Agricultural University , Nanjing , China
Sun Genlou
Electronic publication date: 2024 Aug 7
Publication date: 2024
Volume: 12
Electronic Location ID: e17802
Received 2024 Apr 3; Accepted 2024 Jul 2
Copyright: © 2024 Li-Xiang et al.
Copyright year: 2024
Copyright holder: Li-Xiang et al.
License: This is an open access article distributed under the terms of the Creative Commons Attribution License, which permits unrestricted use, distribution, reproduction and adaptation in any medium and for any purpose provided that it is properly attributed. For attribution, the original author(s), title, publication source (PeerJ) and either DOI or URL of the article must be cited.
License URL: https://creativecommons.org/licenses/by/4.0/

Keywords: Daylily, Hemerocallis, Liliaceae, Flower vegetable, Nutritional quality

Funding: Jiangsu Modern Agriculture (vegetable) Industrial Technology System Shuqian Promotion Demonstration Base JATS2022-324 Priority Academic Program Development of Jiangsu Higher Education Institutions Project (PAPD) The Fruit and Vegetable Breeding and Germplasm Bank Construction of Suqian SQ2022023 The research was supported by the Jiangsu Modern Agriculture (vegetable) Industrial Technology System Shuqian Promotion Demonstration Base Project (JATS2022-324) and the Priority Academic Program Development of Jiangsu Higher Education Institutions Project (PAPD). The Fruit and Vegetable Breeding and Germplasm Bank Construction of Suqian (SQ2022023) supported the APC of this article. The funders had no role in study design, data collection and analysis, decision to publish, or preparation of the manuscript.

==============================
Daylily (Hemerocallis citrina) is a perennial herb of the genus Hemerocallis of Liliaceae. It is also an economically important crop and is widely cultivated. Daylily has nutritional, medicinal and ornamental values. The research literature shows that daylily is a high-quality food raw material rich in soluble sugars, ascorbic acid, flavonoids, dietary fiber, carotenoids, mineral elements, polyphenols and other nutrients, which are effective in clearing heat and diuresis, resolving bruises and stopping bleeding, strengthening the stomach and brain, and reducing serum cholesterol levels. This article reviews the main nutrients of daylily and summarizes the drying process of daylily. In addition, due to the existence of active ingredients, daylily also has a variety of biological activities that are beneficial to human health. This article also highlights the nutritional quality of daylily, the research progress of dried vegetable rehydration technology and dried daylily. In the end, the undeveloped molecular mechanism and functional research status of daylily worldwide are introduced in order to provide reference for the nutritional quality research and dried processing industry of daylily.

Introduction

Daylily (Hemerocallis citrina) also known as golden needle, belongs to the perennial herb of Hemerocallis in Liliaceae family (Hirota et al., 2021; Misiukevičius et al., 2023; Qing et al., 2021; Zhang et al., 2024). Hemerocallis is a unique economic crop with the flower as the main product organ containing protein, vitamins, flavonoids and other rich nutritional elements (Deng et al., 2003; Wu et al., 2021a). Daylily has a long cultivation history and widely planting area in China as vegetable, medicine and ornament, and appears in people’s daily dining table (Duan et al., 2021; Jia et al., 2024).

Daylily prefers warm environments in China (North temperate zone), and the flowering period of daylily is generally from June to August (Figs. 1 and 2). Due to the restriction of high temperature in summer, the physiological metabolism of daylily is relatively strong after picking. It is difficult to store and transport fresh daylily. Short storage period, perishability, browning, bloom prematurely and other phenomena seriously affects the commodity value of daylily (Fig. 3). In order to extend the shelf life and supply period of daylily, dried daylily is the main way of processing and storage of daylily (Fig. 4).

Figure 1 Daylily plants of early spring in green house.

Figure 2 Daylily plants growing in the field.

Figure 3 Harvested fresh daylily.

Figure 4 Dried daylily.

Survey methodology

We searched PubMed (https://pubmed.ncbi.nlm.nih.gov/), the Web of Science (https://webofscience.clarivate.cn/wos/alldb/basic-search), Google Scholar (http://scholar.google.com.hk/), CNKI (https://www.cnki.net/), and Baidu (https://www.baidu.com/) by the searching the following: “daylily/hemerocallis/liliaceae/flower vegetable/nutritional quality” are searched in combination with the subject title or its free term, respectively (the specific searching strategies can be found in the supplemental searching strategies).

Economic value of daylily

Nutritional value of daylily

As a vegetable crop, daylily has various nutritional composition and appropriate nutrition structure with high content of protein, vitamins minerals and low calories (Fig. 5). The proportion of the three major nutrients are: carbohydrates 60%; protein 14%; fat 2%. Daylily is also rich in carbohydrates, vitamin C, carotenoids, mineral elements, polyphenols, fats and amino acids and other nutrients. The carotene content in daylily is several times than that of tomato, and phosphorus content is higher than most other vegetables. It was determined that per 100 g of dried daylily contains 60.1 g carbohydrates, 14.1 g protein, 0.4 g fat, 463 mg calcium, 173 mg phosphorus, 16.5 mg iron, 4.1 mg niacin, 3.44 mg carotene, 0.3 mg thiamine, 0.14 mg riboflavin, etc., (Liu et al., 2020; Qin et al., 2022; Wang et al., 2022a).

Figure 5 Dishes processed with fresh daylily.

The medicinal value of daylily

Daylily has high medicinal value containing gamma-hydroxyglutamic acid, succinic acid, β-sitosterol, aspartate, trehalase and other medicinal ingredients (Liang et al., 2021; Zhan et al., 2005; Zou et al., 2024). Besides the flower buds being used for vegetable, the leaves, roots and seeds of daylily are also valuable with the use as medicine (Fig. 6). Research in the medical field has found that daylily has a sweet and cool taste with good effect in calming the mind and brightening the eyes, strengthening the brain and anti-aging, and can significantly reduce the content of serum cholesterol (Li et al., 2024). Fu & Mao (2008) analyzed the antioxidant properties of extracts from five daylily cultivars, revealing their superior in vitro antioxidant activities and ability to scavenge free radicals. Yan et al. (2023) conducted a detailed study on the phytochemical characteristics and biological functions of steamed daylily, revealing that it is rich in flavonoids and triterpenoids. Additionally, the extract of steamed daylily significantly enhanced the resistance of normal heart and liver cells (H9c2 and L-02 cells) to ethanol damage. Daylily has the effects of enhancing immunity, improving sleep, anti-constipation, invigorating brain and strengthening wisdom, relieving pain, anti-inflammatory, removing stasis and hemostasis (Hsu et al., 2023; Liang et al., 2023a, 2023b). In experiments investigating the effects of daylily on mouse sleep, the extracts from the roots and flowers of daylily were found to reduce spontaneous activity and promote sleep in mice (Uezu, 1998; Li et al., 2024). Recent research confirmed that the aqueous and ethanol extracts of daylily effectively inhibit the production of nitric oxide and interleukin-6 in lipopolysaccharide-activated macrophages (Hsu et al., 2023). In the study, compared to adult volunteers who received only water, those who received the aqueous extract of daylily flowers showed improvements in sleep quality, sleep efficiency, and daytime functioning, with a significant reduction in sleep latency (Hsu et al., 2023). Regular consumption of daylily have the health benefits of anti-bacterial and anti-viral, liver protection, anti-depression, promoting blood circulation and removing blood stasis, and inhibiting tumor (Liu et al., 2022; Ma et al., 2022; Sang, Fu & Song, 2023). The study demonstrated that anthraquinones isolated from daylily flowers significantly inhibit cancer cell proliferation, particularly in lung and breast cancer cells (Cichewicz & Nair, 2002). In the study of the antidepressant effect of extract from daylily, it was found that the extract significantly increased the concentrations of 5-hydroxytryptamine (5-HT) and norepinephrine in the frontal cortex and hippocampus of mice, and enhanced the level of dopamine in the frontal cortex, thus exhibiting potential antidepressant activity (Gu et al., 2012). These findings offered new perspectives on the application of daylily in health and medical fields.

Figure 6 Morphological characteristics of daylily.

Ornamental value of daylily

In addition to its edible and medicinal value, daylily also has a high ornamental value. Daylily was an excellent garden ornamental plant with long flowering period and germination in spring (Hasegawa et al., 2006; Hirota et al., 2012; Kakrana et al., 2018; Wang et al., 2023b, 2023c). Before it was widely cultivated as a vegetable and medicinal crop, daylily plants are used for the decoration of courtyards, meadows and gardens. The leaves of daylily are bright green from spring to late autumn, and they can also be used as ornamental plant in summer, and besides favored as ornamental flowers, it also be used as cut flowers now (Bai, Zhang & Gao, 2023; Guo et al., 2023; Hirota et al., 2012; Ma et al., 2018). Different varieties of daylily with different habits can be cultivated in all year round, so that they can have flowers in all four seasons (Figs. 7 and 8). Cui et al. (2019) conducted a comprehensive study, compiling a collection of 183 Hemerocallis germplasms. Through numerical classification of flower colors, these germplasms were categorized into five distinct groups. This systematic classification facilitates the breeding of a diverse range of daylily cultivars for ornamental purposes, enriching the palette of colors available for appreciation. In another study, a selection of 135 Hemerocallis cultivars, characterized by their abundant blooms, vibrant colors, and distinctive features, was mad (Wang et al., 2022c). Researchers observed and recorded the growth traits of each cultivar throughout a complete growth cycle. Notably, the study identified early-blooming cultivars that flower during the “Mother’s Day” sales peak, a critical period for the ornamental plant market (Wang et al., 2022c). Furthermore, the study highlighted 12 daylily cultivars with evergreen, ability to rebloom (re-flowering), and drought tolerance, which were suitable for production and promotion (Wang et al., 2022c).

Figure 7 The green stems and leaves and bright flowers of daylily.

Figure 8 Bright flowers of daylily.

Main nutrients of daylily

Soluble sugar

Soluble sugar is a kind of water-soluble sugar, which belongs to the effective carbohydrate that can be absorbed and used by humans. Soluble sugar is involved in regulating the sweetness of fruits and vegetables. Soluble sugar is one of the induced small molecular solutes, which can participate in osmoregulation and play an important role in maintaining protein stability (Rosa et al., 2009). In higher plants, the types of soluble sugars mainly include glucose, sucrose, fructose, maltose and trehalose. In order to delay the physiological and metabolic effects caused by drought stress, plants will accumulate small molecular organic soluble carbohydrate compounds in cells. Previous studies on osmoregulation of water stress in sorghum (Sorghum Moench) and maize (Zea mays) seedling stage showed that soluble sugar was involved in osmoregulation and physiological recovery and repair of plants under drought stress after rehydration. It has been reported that the soluble sugar content is closely related to the cold tolerance of plants under low temperature environment (Sami et al., 2016).

The high soluble sugar content is also one of the aspects of the excellent nutritional quality of daylily of high nutritional value and delicious taste. As the measure results showed that the content of soluble sugar in the flower buds of daylily is 69.511 mg·g−1, including 4.216 mg·g−1 sucrose, 25.331 mg·g−1 glucose and 39.964 mg·g−1 fructose, which is at a high level in herb of Hemerocallis relatively (Meng et al., 2021). The soluble sugar content in daylily decreased significantly under high salt or alkali stress treatments (Han et al., 2018). It also decreased significantly with the extension of storage time after harvesting. Appropriate storage methods can effectively delay the decline of soluble sugar content in daylily (Han et al., 2013). In recent research, a novel water-soluble polysaccharide has been isolated from daylily, revealing its potential as an active ingredient for maintaining intestinal health (Ke et al., 2024). It is necessary for further in-depth investigation into the development and comprehensive utilization of soluble sugar in daylilies as functional bioactive substances.

Ascorbic acid

Ascorbic acid (AsA), also known as vitamin C, is a small molecule antioxidant commonly found in higher plant. Ascorbic acid acts as a redox buffer involved in regulating the redox balance of plant cell, regulating gene transcription and protein translation and acting as a coenzyme for some enzymes. The growth and development of higher plants mainly come from the division and elongation of cells. As an antioxidant, natural ascorbic acid is mostly found in fruits and vegetables, involved in the synthesis, elongation and cross-linking of plant cell walls and regulation of cell growth. Ascorbic acid also plays important roles in plant growth and development, and disease resistance and anti-stress defense, and has a certain easing effect on the damage of plants under adverse conditions.

Ascorbic acid is an essential human nutrition element and an important non-enzyme antioxidant in human body. The reaction of living bodies will produce some harmful substances such as oxidation free radicals with strong oxidation, which could damage the body’s tissues and cells, thereby causing some chronic diseases. Ascorbic acid could remove of excess reactive oxygen species and free radicals, and effectively improve human resistance, enhance human immunity. Ascorbic acid has a certain whitening effect, with reduce dark oxymelanin in the process of skin melanin production to reducing melanin, and participate in tyrosine metabolism to reduce the transformation of tyrosine to melanin (Yang, Geng & Fan, 2017). Detected the ascorbic acid content of 61.172 μg·g−1 in daylily flower buds, which is higher than the average level of Hemerocallis plants. Zeng et al. (2024) carried out a comprehensive evaluation on the quality of flower buds from seven daylilies cultivars (‘MZH,’ ‘SYHH,’ ‘CBH,’ ‘QZH,’ ‘DTH,’ ‘MLHH,’ and ‘DWZ’). The findings revealed that all seven daylily cultivars were rich in ascorbic acid with concentrations ranging from 9.10 to 13.45 mg/100 g. Based on the ascorbic acid content, the cultivars were ranked as follows: ‘MLHH’ > ‘DWZ’ > ‘DTH’ > ‘SYHH’ > ‘MZH’ > ‘CBH’ > ‘QZH’ (Zeng et al., 2024). Stress response and storage related researches on daylily showed that ascorbic acid played an antioxidant role under low concentration salt stress (Han et al., 2018). The content of ascorbic acid decreased with the extension of storage time and the optimization of storage conditions could delay the degradation of ascorbic acid in daylily flower buds (Han et al., 2013).

Flavonoids

Flavonoids are a class of secondary metabolites widely existing in the plant kingdom, and they are one of the essential nutrients for human body. Flavonoids are the main effective components of the nutritional and medicinal value of daylily. They have many health effects, such as anti-oxidation, scavenging free radicals, delaying aging, etc., almost no toxic side effects (Yang, Geng & Fan, 2017; Zhan et al., 2005). In higher plant, flavonoids have many biological functions, including regulating plant growth and development, protecting plants from UV damage, and so on. It has been proved that the higher the content of flavonoid extract, the stronger its antioxidant capacity (Feng et al., 2018; Rodríguez-Aguilar, Ortega-Regules & Ramírez-Rodrigues, 2024).

Daylily contains rich flavonoids, the highest content in the leaves. Flavonoids are the main effective components of the nutritional value of daylily (Lv & Guo, 2023; Wang et al., 2022b). They have many functions and little side effects, such as anti-oxidation, scavenging free radicals, delaying aging and so on. Wang et al. (2024) analyzed the nutritional and functional components of daylilies from four different production areas in Shanxi Province using extensive targeted metabolomics approach. The research findings indicated that flavonoids are abundant in daylilies, representing the second most prevalent compound group. Specifically, the researchers identified a total of 287 flavonoids, encompassing various types such as dihydroisoflavones, flavanols, and flavanes. Another recent study integrated transcriptomics, proteomics, and metabolomics to investigate the compositional changes of flavonoids during the development from flower bud to open flower in daylilies. The findings revealed 18 genes differentially expressed flavonoids between the flower bud and open flower stages (one upregulated and 17 downregulated) (Li, Qin & Cui, 2024). This research also identified potential key genes (UGT73C6, CYP81E, CHI and FLS) and transcription factor families (bZIP, MADS, MYB, BHLH) involved in flavonoid metabolism of daylilies (Li, Qin & Cui, 2024). The integration of these multi-omics approaches provides a comprehensive understanding of the molecular mechanisms underlying flavonoid biosynthesis and paves the way for future studies aimed at enhancing the functional components in daylilies.

Dietary fiber

Dietary fiber mainly includes non-starch polysaccharides, cellulose, lignin, pectin and other related substances, known as the “seventh nutrient”. In many areas of China, the proportion of the population with insufficient dietary fiber intake is more than 90%. Increasing dietary fiber intake can improve people’s health. The intake of dietary fiber can slow down many chronic diseases caused by the disorder of glucose and lipid metabolism, including lowering blood sugar, lowering cholesterol, regulating body weight, etc. At the same time, it can also promote the growth and diversity of intestinal microorganisms of human.

The dietary fiber in daylily mainly includes lignin and cellulose. Lignin is a phenolic polymer polymerized by monomers formed by hydroxylation and methylation of phenylalkylamine derivatives. It is a macromolecular organic matter, second only to cellulose in plants, and has important biological functions (Khadr et al., 2020; Que et al., 2019). As a kind of insoluble fiber, lignin can enhance human immunity and improve absorption and digestion functions. Cellulose is the main component of plant cell wall, and is the most widely distributed and the most abundant polysaccharide in nature, accounting for more than 50% of carbon content in the higher plants. The human body does not contain β-glucosidase and can not decompose and use cellulose. Cellulose could absorb a lot of water, promote intestinal peristalsis and hence had a protective effect on the intestine of human. Studies have revealed that the content of dietary fiber in fresh daylily flowers is 1.5 g/100 g, while the content in dried daylily flowers is 6.7 g/100 g (Zhang et al., 2023).

Carotenoids

Carotenoids are a kind of important natural pigments. At present, there are more than 600 kinds of carotenoids with different structures in nature, which are usually divided into two categories: carotene and xanthophyll. Carotenoids are mainly distributed in plastids in higher plant. Carotenoids are important pigments in many flowers and fruits, and their different colors and contents lead to the diversity of plant colors (Wang et al., 2023c, 2023d). Carotenoids in fruits are one of the key factors that determine color and affect the appearance of fruits and vegetable. Carotenoids are also photosynthetic auxiliary pigments and synthetic precursors of ABA and some aromatic substances, and play a decisive role in commercial properties of fruits and vegetable. Vitamin A are the precursors of carotenoids and have strong antioxidant properties. It has an important role in the human body, and can combine with excess reactive oxygen species in the human body to avoid its attack on proteins, DNA, lipids and other biological macromolecules. Carotenoids also enhance immune function and reduce the incidence of various chronic diseases by influencing transcription factors and regulating their downstream targets. In a recent study, the extraction of carotenoids from daylily was assisted by ultrasound, and the total carotenoid content in per 100 g daylily was detected to be 62.16 mg (Wu et al., 2021b). In seven daylilies cultivars (‘MZH,’ ‘SYHH,’ ‘CBH,’ ‘QZH,’ ‘DTH,’ ‘MLHH,’ and ‘DWZ’), the range of β-carotene content was 6.18–10.29 mg/100 g, and the β-carotene content was the highest in ‘CBH’ cultivars (Li, Qin & Cui, 2024).

Mineral elements

Mineral elements mainly refer to the other elements in addition to the four elements of carbon, hydrogen, oxygen and nitrogen, also known as inorganic salts or ash. Mineral elements play an important physiological role in the human body. The human body usually takes in the necessary nutrients from the outside world. Higher plant contained a large number of mineral elements including calcium, magnesium and trace elements including iron, manganese, boron, zinc, copper and molybdenum. Clinical researches have reported that the health efficacy of daylily is closely related to 19 kinds of mineral elements (Zhang et al., 2011).

Mineral elements are necessary for plant growth and development. Based on ICP-MS analysis, the content of potassium, calcium and magnesium was higher, followed by manganese, copper and zinc in daylily (Tang et al., 2017). Li, Qin & Cui (2024) showed that all seven daylily cultivars (‘MZH,’ ‘SYHH,’ ‘CBH,’ ‘QZH,’ ‘DTH,’ ‘MLHH,’ and ‘DWZ’) contained mineral elements such as calcium, magnesium and potassium. In terms of the content of each element, there were significant differences ranging from 3.59 to 4,485.50 mg/kg. Specifically, the contents of calcium, potassium, and magnesium were relatively high, all exceeding 1,609.50 mg/kg. Although the contents of iron and zinc were not as high as the former three, they still occupied a certain proportion (Li, Qin & Cui, 2024). In contrast, the contents of manganese, sodium, and copper are relatively low (Li, Qin & Cui, 2024). It was worth mentioning that the study did not detect harmful heavy metals such as cadmium and arsenic, indicating the safety of daylilies. The detection and comparison of mineral elements content and species in different tissues of daylily showed that the buds and leaves of daylily were the main accumulation sites of mineral elements. The buds of daylily with high mineral elements have no enrichment of contaminated elements in soil. In this point, daylily has good edible safety.

Polyphenols

Polyphenols are one of the important secondary metabolites with antioxidant effects in higher plant, as well as important substrates for enzymatic and non-enzymatic browning. The content of polyphenols had great influence on the nutritional quality and sensory quality of vegetables. A large number of researches have shown that polyphenols play significant roles in antioxidant, anti-aging, scavenging free radicals and other aspects. Their antioxidant function can prevent atherosclerosis, coronary heart disease and other cardiovascular diseases, and can also reduce the threat of high fat on human health. At present, polyphenols have become an important raw material in the research and development of new drugs, and have broad development and application prospects.

Daylily contains rutin, chlorogenic acid and quercetin and other polyphenols. The polyphenol content is high, and the yield of dried product can reach 28% in daylily (Cao & Yang, 2021; Zhou, Gao & Zhang, 2012). The dry-processing and rehydration process of daylily can cause the loss of polyphenols, which leads to the decline of its nutritional quality (Hao et al., 2022). Wang et al. (2024) detected 212 phenolic acids in daylily flowers through metabolomics. Apart from serving as antioxidants, these phenolic acids also exhibit various physiological effects.

Drying processing of daylily

Drying is one of the most commonly used technologies for vegetable and fruit processing. After dehydration and drying, the concentration of soluble substances in vegetables increases to the extent that microorganisms are difficult to utilize. That inhibits the breeding of microorganisms and thus prolonging the storage period of vegetables at room temperature (Liu, Zhang & Hu, 2022). At present, vegetable drying technologies include atmospheric hot air drying, microwave drying, vacuum freeze drying, far infrared drying, osmotic drying and combined drying, etc., Sagar & Suresh (2010). The drying temperature, time and pretreatment of dry products will affect the subsequent rehydration of dry products (Wang et al., 2022a). As a common dried vegetable, drying technology is the main processing method to maintain the nutritional quality of daylily, and drying plays an important role in prolonging the storage time and improving the storage performance and physiological activity of vegetables (Wang et al., 2023a). In addition to the traditional drying process, some new means and methods, such as quick-freezing process, microwave drying and far infrared, have been gradually applied to daylily drying. Previous studies have compared the effects of different drying processes on the drying effect of daylily, and the results show that it is better to choose indoor and outdoor high temperature and low humidity environment. In addition, the drying rate and moisture content of daylily treated by vacuum freeze-drying technology are better (Wu et al., 2021a).

Conclusion and outlook

Daylilies as an important economic crop within the Liliaceae family have a long history of cultivation and consumption in China. Because of its unique flavor and nutritional value, daylilies are favored by consumers. Despite the promotion of large-scale cultivation of daylilies in various regions of China, literature on daylilies is relatively fragmented and lacks systematic organization. This review aims to synthesize the nutritional, medicinal, and ornamental values of daylilies and summarize their key functional components, with the goal of providing a scientific basis for further processing and utilization of daylilies. Additionally, this article also forecasts future research directions and the development trends of the daylily industry.

Drying techniques and processing quality

Daylily products are primarily in the form of dried goods, and different drying methods have a direct impact on the processing speed, quality, and taste of the dried daylilies. Therefore, choosing a suitable drying process could maintain the quality of daylily to the maximum, and meet the needs of industrialization, which has become the focus of daylily processing research. In-depth study of the physiological characteristics and nutritional quality of daylilies is of great significance for improving and perfecting their processing technology.

Development of high value-added products

In terms of the processing and value development of daylilies, there is an urgent need to develop new products with high added value. Moreover, the diversity of the daylily processing industry also needs to be strengthened. It is conducive to the development of high value-added products and the formation of an industrial chain with good economic benefits.

Research and utilization of functional components

Although researchers have made certain progress in the extraction, purification, structural identification, and nutritional activity of functional components of daylilies, there is still a broad space for research. Further research on the functional components of daylily is of great significance for its value development and rational processing and application. At the same time, it provides a new idea for the application of daylily in medicine, health products and other fields.

Ornamental value and industrial application

The ornamental value of daylilies should not be overlooked either. As a beautiful flower, daylilies have a positive role in the construction of “Beautiful Countryside.” In the future, on the basis of industry research, the healthy and high-quality development of the daylily industry should be promoted to produce more high-quality fresh and processed products.

Collection, utilization, and breeding of germplasm resources

Finally, the collection, utilization, and breeding of daylily germplasm resources should also be strengthened. At present, there are many wild varieties of daylilies, but there are bottlenecks in variety breeding, and most varieties rely on introduction. How to effectively use the local regional advantages to accelerate the breeding of new varieties of daylilies will be an important direction for future exploration. We hope that through the development value orientation of daylilies, high-quality and high-yielding daylily varieties will be selected and promoted in a targeted manner to meet market and social needs.

Mining functional genes

Daylily contains a range of bioactive ingredients, such as flavonoids, terpenoids, alkaloids and so on. These ingredients have the beneficial functions of anti-oxidation, anti-tumor, anti-depression, antibacterial and liver protection, and have great application prospects in developing functional products and promoting industrial development. However, the specific components that play a role in daylily need to be further isolated and identified. In addition, genes related to the biosynthesis of bioactive components in daylily have only been predicted, but their specific functions have not been identified.

With the rapid development of sequencing technology, transcriptomics, proteomics, metabolomics and other omics techniques have been widely used in daylily research. The daylily genome has been sequenced and the genome sequence data has been published. The deciphering of the genome of daylily lays a foundation for genetic research and molecular breeding of daylily. The large amount of data generated by these studies provides a potential reference for the mining of functional genes in daylily. Firstly, the identification of key genes can be carried out in combination with a variety of omics methods. Secondly, the establishment of stable genetic transformation system and CRISPR/Cas9 system is an effective way to study gene function and crop improvement of daylily. The mining of functional genes will play a guiding and promoting role in daylily breeding. However, this effective tool has not been widely used in daylily research, and future research should focus on this area.

Additional Information and Declarations

Competing Interests

Author Contributions

Data Availability

The authors declare that they have no competing interests.

Li-Xiang Wang conceived and designed the experiments, performed the experiments, analyzed the data, prepared figures and/or tables, authored or reviewed drafts of the article, and approved the final draft.

Ya-Hui Wang performed the experiments, analyzed the data, prepared figures and/or tables, authored or reviewed drafts of the article, and approved the final draft.

Chen Chen performed the experiments, analyzed the data, prepared figures and/or tables, and approved the final draft.

Jie-Xia Liu performed the experiments, analyzed the data, prepared figures and/or tables, authored or reviewed drafts of the article, and approved the final draft.

Tong Li performed the experiments, analyzed the data, prepared figures and/or tables, authored or reviewed drafts of the article, and approved the final draft.

Jing-Wen Li performed the experiments, analyzed the data, prepared figures and/or tables, authored or reviewed drafts of the article, and approved the final draft.

Pei-Zhuo Liu performed the experiments, analyzed the data, prepared figures and/or tables, and approved the final draft.

De-Bao Xu performed the experiments, analyzed the data, prepared figures and/or tables, and approved the final draft.

Sheng Shu conceived and designed the experiments, prepared figures and/or tables, authored or reviewed drafts of the article, and approved the final draft.

Ai-Sheng Xiong conceived and designed the experiments, prepared figures and/or tables, authored or reviewed drafts of the article, and approved the final draft.

The following information was supplied regarding data availability:

This is a literature review.

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
