# Peer review of "Advances in research on the main nutritional quality of daylily, an important flower vegetable of Liliaceae"

_PeerJ, doi:10.7717/peerj.17802_

## Round 0.1 · original submission · Minor Revisions

Please find the comments from reviewers, and make revisions.

**Language Note:** PeerJ staff have identified that the language needs to be improved. When you prepare your next revision, please either (i) have a colleague who is proficient in English and familiar with the subject matter review your manuscript, or (ii) contact a professional editing service to review your manuscript. PeerJ can provide language editing services - you can contact us at [email protected] for pricing (be sure to provide your manuscript number and title). – PeerJ Staff

Reviewer 1 ·

Basic reporting

1. The section of Conclusion and Outlook can be further expanded, for example, by including prospects for future research directions or predictions on the impact of industrial development, in order to make the entire article more logically complete.
2. The article details the main nutritional components of daylily, including soluble sugars, ascorbic acid, flavonoids, dietary fiber, carotenoids, mineral elements, and polyphenols. These contents are relatively complete, but it may be beneficial to consider incorporating some of the latest research findings or providing a more detailed comparison and analysis of the nutritional differences among different varieties.
3. When describing the medicinal value of daylily, various health benefits such as soothing nerves and improving vision are mentioned. It may be helpful to include some research data supported by literature in this section to enhance the credibility and persuasiveness of the description.
4. Regarding the ornamental value of daylily, the article mentions its application in garden landscaping and flower blooming characteristics. This section can be further expanded, for example, by introducing the ornamental characteristics of different varieties, in order to provide a more comprehensive description of the ornamental value.

Experimental design

no comment

Validity of the findings

no comment

Reviewer 2 ·

Basic reporting

In this manuscript entitled “Advances in research on the main nutritional quality of daylily, an important flower vegetable of Liliaceae” by Wang et al., the author systematically introduced the main nutritional quality of daylily. The application of these main nutritional was also discussed in daylily.
This paper has comprehensive information and detailed content, which has a good guiding effect on understanding and application of daylily. Most parts of this submission are presented satisfactorily, while revisions are still needed.

Experimental design

no comment

Validity of the findings

no comment

Additional comments

Comments:
1. The logic of the abstract needs to be adjusted.
2. Drying technology is the main processing method to maintain the nutritional quality of daylily. It is recommended to add an chapter to introduce this part.
3. It is suggested to add “Dry processing” as a keyword.
4. In the ‘CONCLUSION AND OUTLOOK’ section, could you add a section on the understanding of unexplored Molecular mechanism research in daylily?
5. English needs further polishing.

Reviewer 3 ·

Basic reporting

no comment

Experimental design

no comment

Validity of the findings

no comment

Additional comments

In the manuscript “Advances in research on the main nutritional quality of daylily, an important flower vegetable of Liliaceae”, the authors highlighted the nutritional quality of daylily, the research progress of dried vegetable rehydration technology and dried daylily. The medicinal and ornamental values of daylily were also discussed.
The manuscript provided useful information of daylily. The manuscript could be considered for publication after addressing some points below.

Comments:
1. The article covers many research areas related to daylily research, but the discussion of each aspect is relatively brief. Some only point out the facts, without providing more in-depth and specific information. My advice is that you can provide more specific and enlightening information in addition to the results from other studies.
2. Abstract content is short, it is suggested to increase.
3. It is suggested to add the figure about rehydration of dried daylily in Figure 4.
4. The “CONCLUSION AND OUTLOOK” should be modified to make it more specific and avoid duplication with the abstract.
5. The author should check references carefully. Make sure they are properly cited.
6. The language of the paper also needs to be polished.

---

## Round 0.2 · accepted · Accept

Your manuscript is accepted for publication

Reviewer 1 ·

Basic reporting

no comment

Experimental design

no comment

Validity of the findings

no comment

Additional comments

no comment

Reviewer 2 ·

Basic reporting

The revised manuscript has much improved. All previous questions and suggestions have been addressed.

Experimental design

no comment

Validity of the findings

no comment

Additional comments

no comment

Reviewer 3 ·

Basic reporting

no comment

Experimental design

no comment

Validity of the findings

no comment

Additional comments

The authors have solved most of the issues that I am concerned.